# TOWARDS SAFE DEEP LEARNING: UNSUPERVISED DEFENSE AGAINST GENERIC ADVERSARIAL ATTACKS

## ABSTRACT

Recent advances in adversarial Deep Learning (DL) have opened up a new and largely unexplored surface for malicious attacks jeopardizing the integrity of autonomous DL systems. We introduce a novel automated countermeasure called *Parallel Checkpointing Learners* (*PCL*) to thwart the potential adversarial attacks and significantly improve the *reliability* (safety) of a victim DL model. The proposed PCL methodology is *unsupervised*, meaning that no adversarial sample is leveraged to build/train parallel checkpointing learners. We formalize the goal of preventing adversarial attacks as an optimization problem to minimize the *rarely observed regions* in the latent feature space spanned by a DL network. To solve the aforementioned minimization problem, a set of complementary but disjoint checkpointing modules are trained and leveraged to validate the victim model execution in parallel. Each checkpointing learner explicitly characterizes the geometry of the input data and the corresponding high-level data abstractions within a particular DL layer. As such, the adversary is required to *simultaneously* deceive all the defender modules in order to succeed. We extensively evaluate the performance of the PCL methodology against the state-of-the-art attack scenarios, including Fast-Gradient-Sign (FGS), Jacobian Saliency Map Attack (JSMA), Deepfool, and Carlini&WagnerL2. Extensive proof-of-concept evaluations for analyzing various data collections including MNIST, CIFAR10, and ImageNet corroborate the effectiveness of our proposed defense mechanism against adversarial samples.

## 1 INTRODUCTION

Security and safety consideration is a major obstacle to the wide-scale adoption of emerging learning algorithms in sensitive scenarios, such as intelligent transportation, healthcare, and video surveillance applications (McDaniel et al. (2016); Dahl et al. (2013); Knorr (2015)). While advanced learning technologies are essential for enabling coordination and interaction between the autonomous agents and the environment, a careful analysis of their decision reliability in the face of carefully crafted adversarial samples (Goodfellow et al. (2014); Papernot et al. (2016a); Moosavi-Dezfooli et al. (2016); Carlini & Wagner (2017b)) and thwarting their vulnerabilities are still in their infancy.

Consider a traffic sign classifier employed in self-driving cars. In this setting, an adversary can carefully add imperceptible perturbation to a legitimate "stop" sign sample and fool the DL model to classify it as a "yield" sign; thus, jeopardizes the safety of the vehicle as shown in (McDaniel et al. (2016)). As such, it is highly important to reject risky adversarial samples to ensure the integrity of DL models used in autonomous systems such as unmanned vehicles and drones. In this paper, we aim to answer two open questions regarding the adversarial attacks.

*(i) Why are machine learning models vulnerable to adversarial samples?* Our hypothesis is that the vulnerability of neural networks to adversarial samples originates from the existence of rarely explored sub-spaces in each feature map. This phenomenon is particularly caused by the limited access to the labeled data and/or inefficiency of regularization algorithms (Wang et al. (2016); Denil et al. (2013)). Figure 1 provides a simple illustration of the partially explored space in a two-dimensional setup. We analytically and empirically back up our hypothesis by extensive evaluations on the state-of-the-art attacks, including Fast-Gradient-Sign (Goodfellow et al. (2014)), Jacobian Saliency Map Attack (Papernot et al. (2016a)), Deepfool (Moosavi-Dezfooli et al. (2016)), and Carlini&WagnerL2 (Carlini & Wagner (2017b)).

*(ii) How can we characterize and thwart the underlying space for unsupervised model assurance as well as defend against the adversaries?* A line of research has shown that there is a trade-off

between the robustness of a model and its accuracy (Madry et al. (2017); Papernot et al. (2016b)). Taking this into account, instead of making a single model that is both robust and accurate, we introduce a new defense mechanism called Parallel Checkpointing Learners (PCL). In this setting, the victim model is kept *as is* while separate defender modules are trained to *checkpoint* the data abstractions and assess the reliability of the victim's prediction. Each defender module characterizes the explored sub-space in the pertinent layer by learning the probability density function (pdf) of legitimate data points and marking the complement sub-spaces as rarely observed regions. Once such characterization is obtained, the checkpointing modules[1] evaluate the input sample in parallel with the victim model and raise alarm flags for data points that lie within the rarely explored regions (Figure 1c). As we demonstrate in Section 4, adversarial samples created by various attack methods mostly lie within the sub-spaces marked as partially explored sectors.

We consider a *white-box attack model* in which the attacker knows everything about the victim model including its architecture, learning algorithm, and parameters. This threat model represents the most powerful attacker that can endanger the real-world applications. We validate the security of our proposed approach for different DL benchmarks including MNIST, CIFAR10, and a subset of ImageNet data. Based on the result of our analysis, we provide new insights on the reason behind the existence of adversarial transferability. We open-source our API to ensure ease of use by the users (the link is omitted for blind review purposes) and invite the community to attempt attacks against our provided benchmarks in the form of a challenge.

The explicit contribution of this paper is as follows: **(i)** Devising an automated end-to-end framework for unsupervised model assurance as well as defending against the adversaries. **(ii)** Incepting the idea of parallel checkpointing learners to validate the legitimacy of data abstractions at each intermediate DL layer. **(iii)** Performing extensive proof-of-concept evaluations against state-of-the-art attack methods. **(iv)** Providing new insights regarding the transferability of adversarial samples in between different models.

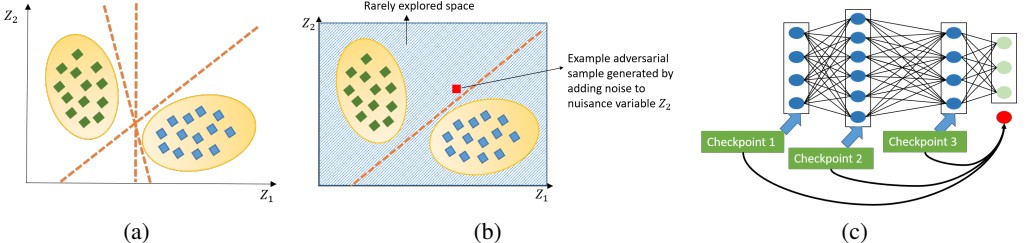

(a)  (b)  (c)

Figure 1: (a) In this example, data points (denoted by green and blue squares) can be easily separated in one-dimensional space. Having extra dimensions adds ambiguity in choosing the pertinent decision boundaries. For instance, all the shown boundaries (dashed lines) are sufficient to classify the raw data with full accuracy in two-dimensional space but are not equivalent in terms of robustness to noise. (b) The rarely explored space (region specified by diagonal striped) in a learning model leaves room for adversaries to manipulate the nuisance (non-critical) variables and mislead the model by crossing the decision boundaries. (c) In PCL methodology, a set of defender (checkpoint) modules is trained to characterize the data density distribution in the space spanned by the victim model. The defender modules are then used in parallel to checkpoint the reliability of the ultimate prediction and raise an alarm flag for risky samples.

## 2 TRAINING CHECKPOINTING MODULES FOR INTERMEDIATE LAYERS

The goal of each defender (checkpointing) module is to learn the pdf of the explored sub-spaces in a particular intermediate DL feature map. The learned density function is then used to identify the rarely observed regions as depicted in Figure 1b. We consider a Gaussian Mixture Model (GMM) as the prior probability to characterize the data distribution at each checkpoint location.[2]

---

[1]We use the term "*checkpointing module*" and "*defender module*" interchangeably throughout the paper.

[2]It is worth noting that our proposed approach is rather generic and is not restricted to the GMM distribution. The GMM distribution can be replaced with any other prior depending on the application.

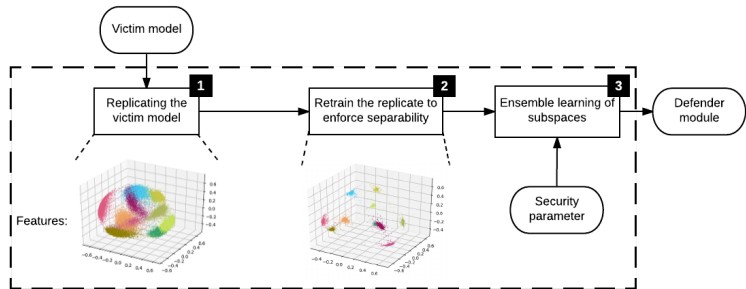

Figure 2: Block diagram of the training procedure for devising parallel checkpointing modules. Each defender module is built by minimizing the disentanglement of intermediate data features in a Euclidean space at a particular checkpoint location. This goal is achieved through several rounds of iterative realignment of data abstractions. The latent data space is then characterized as an ensemble of lower dimensional sub-spaces to effectively learn the pdf of explored regions and detect atypical samples based on a user-defined security parameter.

To effectively characterize the explored sub-space as a GMM distribution, one is required to minimize the entanglement between each two Gaussian distribution (corresponding to every two different classes) while decreasing the inner-class diversity. Figure 2 illustrates the high-level block diagram of the training procedure for devising a parallel checkpointing module. Training a defender module is a one-time offline process and is performed in three steps.

**1** Replicating the victim neural network and all its feature maps. An $L_2$ normalization layer is inserted in the desired checkpoint location. The normalization layer maps the latent feature variables, $\phi(x)$, into the Euclidean space such that the acquired data embeddings live in a d-dimensional hypersphere, i.e., $\|\phi(x)\|_2 = 1$. This normalization is crucial as it partially removes the effect of over-fitting to particular data samples that are highly correlated with the underlying DL parameters.[3]

**2** Fine-tuning the replicated network to enforce disentanglement of data features (at a particular checkpoint location). To do so, we optimize the defender module by incorporating the following loss function with the conventional cross entropy loss:

$$\mathscr{L} += \gamma \, [ \, \underbrace{\|C^{y^*} - \phi(x)\|_2^2}_{loss_1} \, - \, \underbrace{\Sigma_{i \neq y^*}\|C^i - \phi(x)\|_2^2}_{loss_2} \, + \, \underbrace{\Sigma_i(\|C^i\|_2 - 1)^2}_{loss_3} \, ]. \tag{1}$$

Here, $\gamma$ is a trade-off parameter that specifies the contribution of the additive loss term, $\phi(x)$ is the corresponding feature vector of input sample $x$ at the checkpoint location, $y^*$ is the ground-truth label, and $C^i$ denotes the center of all data abstractions ($\phi(x)$) corresponding to class $i$. The center values $C^i$ and intermediate feature vectors $\phi(x)$ are trainable variables that are learned by fine-tuning the defender module. In our experiments, we set the parameter $\gamma$ to 0.01 and retrain the defender model with the same optimizer used for training the victim model. The learning rate of the optimizer is set to $\frac{1}{10}$ of that of the victim model as the model is already in a relatively good local minima.

Figure 3a illustrates the optimization goal of each defender module per Eq. (1). The first term ($loss_1$) in Eq. (1) aims to condense latent data features $\phi(x)$ that belong to the same class. Reducing the inner-class diversity, in turn, yields a sharper Gaussian distribution per class. The second term ($loss_2$) intends to increase the intra-class distance between different categories and promote separability. The composition of the first two terms in Eq. (1) can be arbitrarily small by pushing the centers to ($C^i \leftarrow \pm\infty$). We add the term, $loss_3$, to ensure that the underlying centers lie on a unit d-dimensional hyper-sphere and avoid divergence in training the defender modules.

Figures 3b and 3c demonstrate the distance of legitimate (blue) and adversarial (red) samples from the corresponding centers $C^i$ in a checkpoint module before and after retraining.[4] As shown, fine-tuning the defender module with proposed objective function can effectively separate the distribution

---

[3]The $L_2$ norm is selected to be consistent with our assumption of GMM prior distribution. This norm can be easily replaced by an arbitrarily user-defined norm through our accompanying API.

[4]The centers $C^i$ before fine-tuning the checkpoint (defender) module are equivalent to the mean of the data points in each class.

of legitimate samples from malicious data points. Note that training the defender module is carried out in an unsupervised setting, meaning that no adversarial sample is included in the training phase.

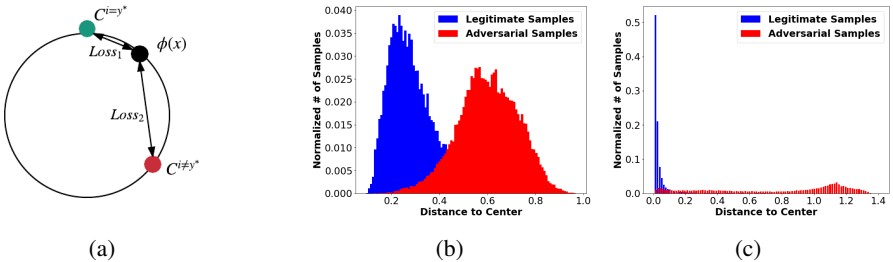

(a)  (b)  (c)

Figure 3: (a) Illustration of the optimization objective in each defender module. (b) The distance of legitimate (blue) and adversarial (red) samples from the corresponding centers $C^i$ before, and (c) after realignment of data samples. In this example, we consider the LeNet3 model (LeCun et al. (1998a)) trained on MNIST dataset (the checkpoint is inserted in the second-to-last layer) and adversarial samples are generated by FGS attack with different perturbation levels.

**3** High dimensional real-world datasets can be represented as an ensemble of lower dimensional sub-spaces (Bouveyron et al. (2007); Mirhoseini et al. (2016); Rouhani et al. (2017)). As discussed in (Bouveyron et al. (2007)), under a GMM distribution assumption, the data points belonging to each class can be characterized as a spherical density in two sub-spaces: (i) The sub-space where the data actually lives ($E_i$) and (ii) its orthogonal complementary space ($E_i^\perp$). The orthogonal space ($E_i^\perp$) is defined such that $E_i^\perp \bigoplus E_i = R^d$, where d is the overall dimensionality of the underlying space. We leverage High Dimensional Discriminant Analysis (HDDA) algorithm (Bouveyron et al. (2007)) to learn the mean and the conditional covariance of each class as a composition of lower dimensional sub-spaces. Under the Gaussian distribution and our specific assumptions, the conditional covariance matrix contains two different eigenvalues $a_i > b_i$ to be determined as shown in (Bouveyron et al. (2007)).

The learned pdf variables (i.e., mean and conditional covariance) are used to compute the probability of a feature point $\phi(x)$ coming from a specific class. In particular, for each incoming test sample $x$, the probability $p(\phi(x)|y^i)$ is evaluated where $y^i$ is the predicted class (output of the victim neural network) and $\phi(x)$ is the corresponding data abstraction at the checkpoint location. The acquired likelihood is then compared against a user-defined *cut-off threshold* which we refer to as the *security parameter*. The Security Parameter (SP) is a constant number in the range of $[0\% - 100\%]$ that determines the hardness of defender modules. Figure 4 illustrates how the SP can control the hardness of the pertinent decision boundaries. In this example, we have depicted the latent features of one category that are projected into the first two Principal Component Analysis (PCA) components in the Euclidean space (each point corresponds to a single input image). The blue and black contours correspond to security parameters of 10% and 20%, respectively. For example, 10% of the legitimate training samples lie outside the contour specified with $SP = 10\%$.

One may speculate that an adversary can add a structured noise to a legitimate sample such that the data point is moved from one cluster to the center of the other clusters; thus fooling the defender modules (Figure 5a). The risk of such attack approach is significantly reduced in our proposed PCL countermeasure due to three main reasons: (i) Use of parallel checkpointing modules; the attacker requires to simultaneously deceive all the defender models in order to succeed. (ii) Increasing intra-class distances in each checkpointing module; The latent defender modules are trained such that not only the inner-class diversity is decreased, but also the distance between each pair of different classes is increased (see Eq. (1)). (iii) Learning a separate defender module in the input space to validate the Peak Signal-to-Noise Ratio (PSNR) level of the incoming samples as discussed in Section 3. In the remainder of the paper, we refer to the defender modules operating on the input space as the *input defenders*. PCL modules that checkpoint the intermediate data features within the DL network are referred as *latent defenders*.

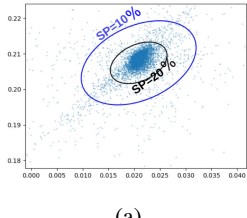 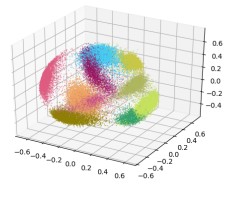 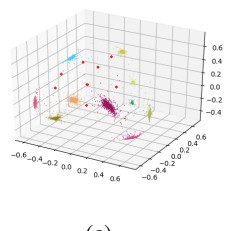

|(a)|(b)|(c)|

Figure 4: (a) Illustration of the effect of security parameter (SP) on the detection policy. A high SP leads to a tight boundary which treats most samples as adversarial examples. (b) Example feature samples in the second-to-last layer of LeNet3 trained for classifying MNIST data. The three axis show the first three Eigenvectors corresponding to the PCA of the data. The first three dimensions are used for visualization purposes only, whereas, the actual data points belong to higher dimensional spaces. (c) Latent feature samples of the same layer in the defender module after data realignment. The majority of adversarial samples (e.g., the red dot points) reside in the regions with low density of training samples.

## 2.1 RISK ANALYSIS

Detecting malicious samples can be cast as a two-category classification task. Let us refer to the category of the legitimate samples as $W_1$ and the category of adversarial samples as $W_2$. If we define $\eta_{ij} = \eta(\alpha_i|W_j)$ as the misclassification penalty[5] incurred for deciding $W_i$ when the true state is $W_j$, the conditional risk in each of our checkpointing modules is equal to:

$$
\begin{aligned}
\mathscr{R}(\alpha_1|\phi(x)) &= \eta_{11}P(W_1|\phi(x)) + \eta_{12}P(W_2|\phi(x)), \\
\mathscr{R}(\alpha_2|\phi(x)) &= \eta_{21}P(W_1|\phi(x)) + \eta_{22}P(W_2|\phi(x)).
\end{aligned}
\tag{2}
$$

The fundamental rule to express the minimum-risk decision is to decide $W_1$ if $\mathscr{R}(\alpha_1|\phi(x)) < \mathscr{R}(\alpha_2|\phi(x))$. In terms of the posterior probabilities, we decide $W_1$ if:

$$
(\eta_{21} - \eta_{11})P(W_1|\phi(x)) > (\eta_{12} - \eta_{22})P(W_2|\phi(x)).
\tag{3}
$$

Generally speaking, the penalty incurred for making an error is greater than the cost incurred for being correct; thus both of the terms $\eta_{21} - \eta_{11}$ and $\eta_{12} - \eta_{22}$ are positive. Following the Bayes' rule, we should select a sample as a legitimate one ($W_1$) if:

$$
(\eta_{21} - \eta_{11})P(\phi(x)|W_1)P(W_1) > (\eta_{12} - \eta_{22})P(\phi(x)|W_2)P(W_2),
\tag{4}
$$

and select $W_2$ otherwise. By reordering the aforementioned decision criteria we have:

$$
\frac{P(\phi(x)|W_1)}{P(\phi(x)|W_2)} > \frac{(\eta_{12} - \eta_{22})}{(\eta_{21} - \eta_{11})} \frac{P(W_2)}{P(W_1)}.
\tag{5}
$$

Note that the right-hand term in Eq. (5) is application specific and is independent of the input data observation $\phi(x)$. In other words, the optimal decision criteria particularly rely on the cost of making a mistake in the given task and the risk of being attacked. This term is tightly correlated with the user-defined cut-off threshold (security parameter) depicted in Figure 4.

Under the GMM assumption, the conditional probability $P(\phi(x)|W_1)$ in Eq. (5) is computed as:

$$
p(\phi(x)|y^i) = \frac{1}{(2\pi)^{\frac{N}{2}}|\Sigma_i|^{\frac{1}{2}}} exp\{-\frac{1}{2}(\phi(x) - \mu_i)^T\Sigma_i^{-1}(\phi(x) - \mu_i)\},
\tag{6}
$$

where $y^i$ is the output of the victim neural network (predicted class), $\mu_i$ and $\Sigma_i$ are the output of the HDDA analysis, and $N$ is the dimension of the latent feature space in the checkpoint module.

---

[5]The misclassification penalty is a constant value which determines the cost of each decision.

## 3 TRAINING CHECKPOINTING MODULES FOR THE INPUT SPACE

We leverage dictionary learning and sparse signal recovery techniques to measure the PSNR of each incoming sample and automatically filter out atypical samples in the input space. Figure 5b illustrates the high-level block diagram of an input defender module. As shown, devising an input checkpoint model is performed in two main steps: (i) dictionary learning, and (ii) characterizing the typical PSNR per class after sparse recovery.

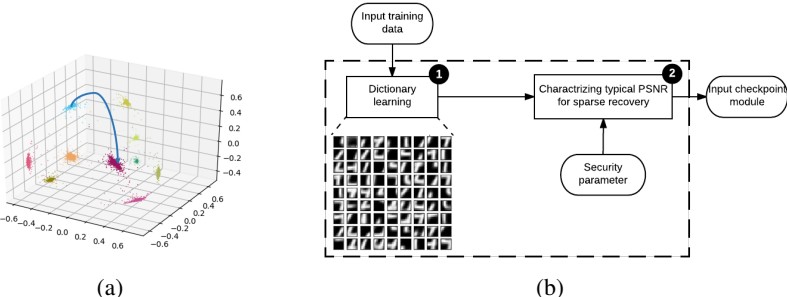

(a)                                                      (b)

Figure 5: An input defender module is devised based on robust dictionary learning techniques to automatically filter out test samples that highly deviate from the typical PSNR of data points within the corresponding predicted class (output of victim model).

**❶** Dictionary learning; we learn a separate dictionary for each class of data by solving:

$$\underset{D^i}{argmin}\, \frac{1}{2}\|Z^i - D^i V^i\|_2^2 \,+\, \beta\|V^i\|_1 \ \ s.t. \ \|D_k^i\|= 1,\, 0 \le k \le k_{max}. \tag{7}$$

Here, $Z^i$ is a matrix whose columns are pixels extracted from different regions of input images belonging to category $i$. For instance, if we consider $8 \times 8$ patches of pixels, each column of $Z^i$ would be a vector of 64 elements. The goal of dictionary learning is to find matrix $D^i$ that best represents the distribution of pixel patches from images belonging to class $i$. We denote the number of columns in $D^i$ by $k_{max}$. For a certain $D^i$, the image patches $Z^i$ are represented with a sparse matrix $V^i$, and $D^i V^i$ is the reconstructed patches. We leverage Least Angle Regression (LAR) method to solve the Lasso problem defined in Eq. (7). In our experiments, we learn a dictionary of size $k_{max} = 225$ for each class of data points using 150,000 randomly selected patches of training data.

For an incoming sample, during the execution phase, the input defender module takes the output of the victim DL model (e.g., predicted class $i$) and uses Orthogonal Matching Pursuit (OMP) routine (Tropp et al. (2007)) to sparsely reconstruct the input data with the corresponding dictionary $D^i$. The dictionary matrix $D^i$ contains a set of samples that commonly appear in the training data belonging to class $i$; As such, the input sample classified as class $i$ should be well-reconstructed as $D^i V^*$ with a high PSNR value, where $V^*$ is the optimal solution obtained by the OMP routine. During the execution phase, all of the non-overlapping patches within the image are denoised by the dictionary to form the reconstructed image.

**❷** Characterizing typical PSNR in each category; we profile the PSNR of legitimate samples within each class and find a threshold that covers all legitimate training samples. If an incoming sample has a PSNR lower than the threshold (i.e., high perturbation after reconstruction by the corresponding dictionary), it will be regarded as a malicious data point. In particular, PSNR is defined as:

$$PSNR = 20log_{10}(MAX_I) - 10log_{10}(MSE), \tag{8}$$

where the mean square error (MSE) is defined as the $L_2$ difference of the input image and the reconstructed image based on the corresponding dictionary. The $MAX_I$ term is the maximum possible pixel value of the image (usually equivalent to 255).

Figure 6 demonstrates the impact of perturbation level on the pertinent adversarial detection rate for three different security parameters (cut-off thresholds). In this experiment, we have considered the FGS attack with different $\varepsilon$ values on the MNIST benchmark.[6] As shown, the use of input dictionaries facilitate automated detection of adversarial samples with relatively high perturbation (e.g.,

---

[6]Table 2 in Appendix A summarizes the DL model topology used in each benchmark. The latent defender module (checkpoint) is inserted at the second-to-last layers.

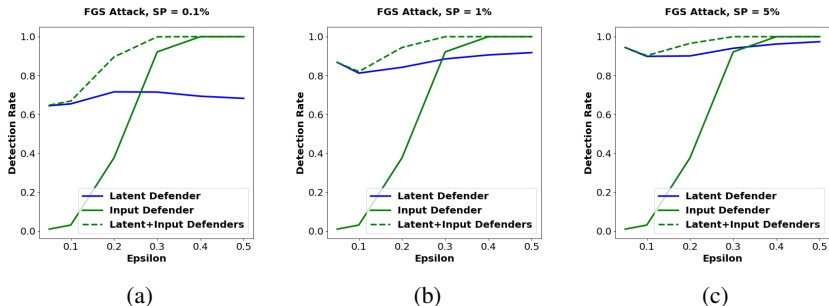

(a)            (b)            (c)

Figure 6: Adversarial detection rate of the latent and input defender modules as a function of the perturbation level for (a) $SP = 0.1\%$, (b) $SP = 1\%$, and (c) $SP = 5\%$. In this experiment, the FGS attack is used to generate adversarial samples and the perturbation is adjusted by changing its specific attack parameter $\varepsilon$.

$\varepsilon > 0.25$) while the latent defender module is sufficient to effectively distinguish malicious samples even with very small perturbations. We extensively evaluate the impact of security parameter on the ultimate system performance for various benchmarks in Section 4.

## 4 EXPERIMENTS

We evaluate the proposed PCL methodology on three canonical machine learning datasets: MNIST (LeCun et al. (1998b)), CIFAR10 (Krizhevsky & Hinton (2009)), and a sub-set of ImageNet (Deng et al. (2009)) consisting of 10 different classes. A detailed summary of the neural network architectures used in each benchmark along with the specific parameters used for various attacks are provided in Appendix A. We leveraged the attack benchmark sets available at (Nicolas Papernot (2017)) for evaluation of different state-of-the-art attacks including FGS, JSMA, Deepfool, and Carlini&WagnerL2 attacks.

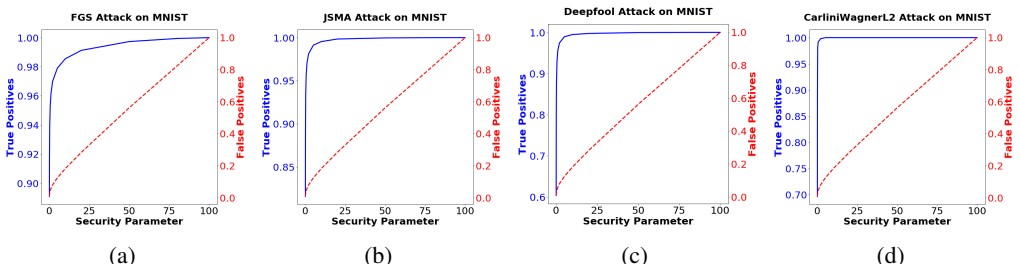

(a)            (b)            (c)            (d)

Figure 7: Impact of security parameter on the ultimate performance of MRR module. The false positive rate is defined as the ratio of legitimate test samples that are mistaken for adversarial samples by the defender modules. The true positive rate is defined as the ratio of adversarial samples correctly classified as malicious data point over the total number of malicious samples. Note that the scales for false positive and true positive axis are different. The false positive rate is computed by considering legitimate samples that are correctly classified by the victim model.

In our proposed countermeasure, the input and latent defenders are jointly considered to detect adversarial samples. In particular, we treat an input as an adversarial sample if either of the latent or input checkpointing modules raise an alarm signal. Figure 7 demonstrates the impact of security parameter on the ultimate false positive and true positive rates for the MNIST benchmark. As shown, a higher security parameter results in a higher true positive detection rate while it also increases the risk of labeling legitimate samples as possibly malicious ones.

To consider the joint decision metric for each application and attack model, we evaluate the false positive and true positive rates and present the pertinent Receiver Operating Characteristic (ROC) curves in Figure 8. The ROC curves are established as follows: first, we consider a latent defender and change the security parameter (SP) in the range of $[0\% - 100\%]$ and evaluate the FP and TP rates for each security parameter, which gives us the dashed blue ROC curves. Next, we consider an input

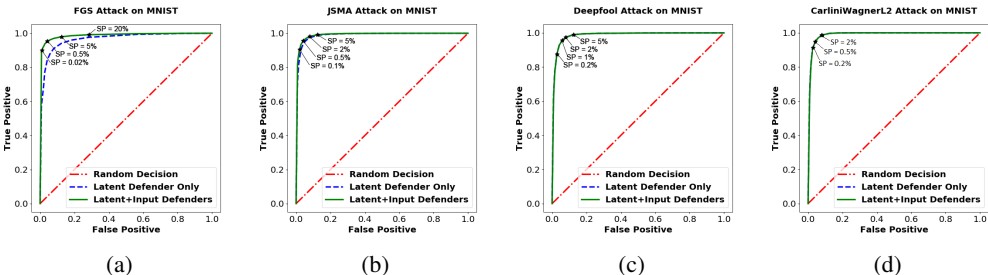

(a)           (b)           (c)           (d)

Figure 8: ROC performance curve of PCL methodology against FGS, JSMA, Deepfool, and Carlini&WagnerL2 attacks. The diagonal line indicates the trajectory obtained by a random prediction.

defender and modify the detection policy: a sample is considered to be malicious if either of the input or latent defenders raise an alarm flag. The ROC curve for this joint defense policy is shown as the green curves in Figure 8. The gap between the dashed blue curve and the green curve indicates the effect of the input defender on the overall decision policy; as can be seen, the input defender has more impact for the FGS attack. This is compatible with our intuition since, compared to the other three attack methods, the FGS algorithm induces more perturbation to generate adversarial samples.

Table 1: PCL performance against different attack methodologies for MNIST, CIFAR10, and ImageNet benchmarks. The reported numbers correspond to the pertinent false positives for achieving particular detection rates in each scenario. The JSMA attack for the ImageNet benchmark is computationally expensive (e.g., it took more than 20min to generate one adversarial sample on an NVIDIA TITAN Xp GPU). As such, we could not generate the adversarial samples of this attack using the JSMA library provide by (Nicolas Papernot (2017)).

| Benchmark | MNIST | | | | CIFAR10 | | | | ImageNet | | | |
|---|---|---|---|---|---|---|---|---|---|---|---|---|
| Detection Rate      Attack | 90% | 95% | 98% | 99% | 90% | 95% | 98% | 99% | 90% | 95% | 98% | 99% |
| FGS | 1.1% | 4.2% | 12.4% | 2.84% | 8.1% | 21.1% | 62.9% | 62.9% | 14.2% | 26.8% | 60.7% | 60.7% |
| JSMA | 2.1% | 4.2% | 8.0% | 12.4% | 8.1% | 14.9% | 21.1% | 33.0% | - | - | - | - |
| Deepfool | 2.8% | 5.9% | 8.0% | 12.4% | 12.0% | 17.9% | 33.0% | 40.8% | 8.1% | 8.1% | 14.2% | 21.5% |
| Carlini&WagnerL2 | 2.8% | 4.2% | 8.0% | 8.0% | 12.0% | 17.9% | 33.1% | 40.8% | 7.9% | 7.9% | 14.0% | 21.3% |

We summarize the performance of the PCL methodology against each of the FGS, JSMA, Deepfool, and Carlini&WagnerL2 attacks for MNIST, CIFAR10, and ImageNet in Table 1. The reported numbers in this table are gathered as follows: we consider a few points on the green ROC curve (marked on Figure 8), which correspond to certain TP rates (i.e., 90%, 95%, 98%, and 99%), then report the FP rates for these points. In all our experiments, the use of only one latent defender module to checkpoint the second-to-last layer of the pertinent victim model was enough to prevent adversarial samples generated by the existing state-of-the-art attacks. Please refer to Appendix B for the complete set of ROC curves for the CIFAR10 and ImageNet benchmarks.

## 5    DISCUSSION

Figure 9 demonstrates an example of the adversarial confusion matrices for victim neural networks with and without using parallel checkpointing learners. In this example, we set the security parameter to only 1%. As shown, the adversarial sample generated for the victim model **are not transferred** to the checkpointing modules. In fact, the proposed PCL approach can effectively remove/detect adversarial samples by characterizing the rarely explored sub-spaces and looking into the statistical density of data points in the pertinent space.

Note that the remaining adversarial samples that are not detected in this experiment are crafted from legitimate samples that are inherently hard to classify even by a human observer due to the closeness of decision boundaries corresponding to such classes. For instance, in the MNIST application, such adversarial samples mostly belong to class 5 that is misclassified to class 3 or class 4 misclassified as 9. Such misclassifications are indeed the model approximation error which is well-understood to the statistical nature of the models. As such, a more precise definition of adversarial samples

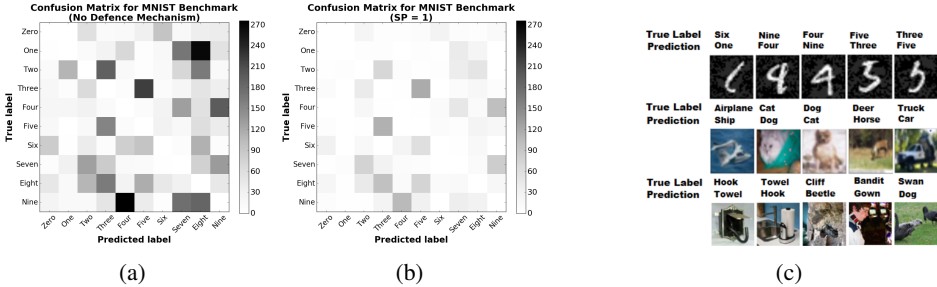

(a)        (b)        (c)

Figure 9: Example adversarial confusion matrix (a) without PCL defense mechanism, and (b) with PCL defense and a security parameter of (1%). (c) Example adversarial samples for which accurate detection is hard due to the closeness of decision boundaries for the corresponding classes.

is extremely required to distinguish malicious samples form those that simply lie near the decision boundaries.

We emphasize that the PCL defenders are trained in an unsupervised setting independent of the attack strategy, meaning that no adversarial sample is used to train the defender models. This is particularly important as it corroborates the effectiveness of the proposed countermeasure in the face of generic attack scenarios including possible future adversarial DL algorithms. Nevertheless, one might question the effectiveness of the proposed approach for adaptive attack algorithms that target the defender modules. A comprehensive study of possible adaptive attack algorithms is yet to be performed if such attacks are developed in the future. We emphasize that, thus far, we have been able to significantly thwart all the existing attacks with only one checkpoint model approximating the data distribution in the second-to-last layer of the corresponding models. Our proposed PCL methodology, however, provides a rather more generic approach that can be adapted/modified against potential future attacks by training parallel disjoint models (with diverse objectives/parameters) to further strengthen the defense.

Figure 10 demonstrate how using multiple checkpoints with a negative correlation in parallel can effectively reduce the number of false alarms while increasing the detection rate of adversarial samples. In this experiment, we have considered MNIST data classification using LeNet model with 4 layers and FGS attack. The checkpoints are inserted in different layers of the pertinent neural network (first layer up to the second-to-last layer). We empirically select the mixing coefficients to aggregate the confidence of the checkpoint defenders for rejecting an incoming sample.

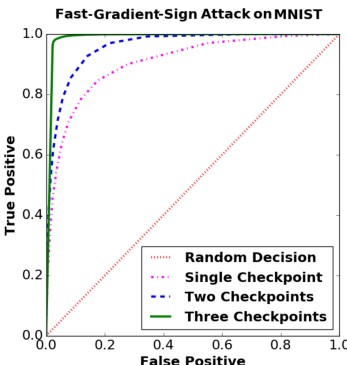

Figure 10: Leveraging multiple parallel checkpoint modules can significantly improves the DL model prediction reliability while minimizing the number of false alarms.

Note that, there is a trade-off between the computational complexity (e.g., runtime overhead) of the PCL defenders and the reliability of the overall system. On the one hand, a high number of validation checkpoints increases the reliability of the systems, but it also increases the computational load as each input sample should be validated by more defender networks. On the other hand, a small number of checkpoints may degrade the defense mechanism performance by treating adversarial samples as legitimate ones. We are looking into automated techniques to customize the number

of checkpoint modules and their corresponding mixing coefficients based on application data and physical constraints such as real-time analysis requirement as future work.

## 6 RELATED WORK

In response to the various adversarial attack methodologies proposed in the literature (e.g., Goodfellow et al. (2014); Papernot et al. (2016a); Moosavi-Dezfooli et al. (2016); Carlini & Wagner (2017b)), several research attempts have been made to design DL strategies that are more robust in the face of adversarial examples. The existing countermeasures can be classified into two distinct categories:

**(i)** Supervised strategies which aim to improve the generalization of the learning models by incorporating the noise-corrupted version of inputs as training samples (Jin et al. (2015); Gu & Rigazio (2014)) and/or injecting adversarial examples generated by different attacks into the DL training phase (Huang et al. (2015); Shaham et al. (2015); Goodfellow et al. (2014); Szegedy et al. (2013)). The proposed defense mechanisms in this category are particularly tailored for specific perturbation patterns and can only partially evade adversarial samples generated by other attack scenarios (with different perturbation distributions) from being effective as shown in (Gu & Rigazio (2014)).

**(ii)** Unsupervised approaches which aim to smooth out the underlying gradient space (decision boundaries) by incorporating a smoothness penalty (Miyato et al. (2015); Carlini & Wagner (2017b)) as a regularization term in the loss function or compressing the neural network by removing the nuisance variables (Papernot et al. (2016b)). These set of works have been mainly remained oblivious to the pertinent data density in the latent space. In particular, these works have been developed based on an implicit assumption that the existence of adversarial samples is due to the piece-wise linear behavior of decision boundaries (obtained by gradient descent) in the high-dimensional space. As such, their integrity can be jeopardized by considering different perturbations at the input space and evaluating the same attack on various perturbed data points to even pass the smoothed decision boundaries as shown in (Carlini & Wagner (2016)). More recently, Meng & Chen (2017) propose an unsupervised manifold projection method called MagNet to reform adversarial samples using autoencoders. Unlike PCL, MagNet is inattentive to the density function of the data in the space. As shown in Carlini & Wagner (2017a), manifold projection methods including MagNet are not robust to adversarial samples and can approximately increase the required distortion to generate adversarial sample by only 30%.

To the best of our knowledge, the proposed PCL methodology is the first unsupervised countermeasure developed based upon probabilistic density analysis and dictionary learning to effectively characterize and thwart adversarial samples. The PCL method does not assume any particular attack strategy and/or perturbation pattern. This is particularly important as it demonstrates the generalizability of the proposed approach in the face of adversarial attacks.

## 7 CONCLUSION

This paper proposes a novel end-to-end methodology for characterizing and thwarting adversarial DL space. We introduce the concept of parallel checkpointing learners as a viable countermeasure to significantly reduce the risk of integrity attacks. The proposed PCL methodology explicitly characterizes statistical properties of the features within different layers of a neural network by learning a set of complementary dictionaries and corresponding probability density functions. The effectiveness of the PCL approach is evaluated against the state-of-the-art attack models including FGS, JSMA, Deepfool, and Carlini&WagnerL2. Proof-of-concept experiments for analyzing various data collections including MNIST, CIFAR10, and a subset of the ImageNet dataset corroborate successful detection of adversarial samples with relatively small false-positive rates. We devise an open-source API for the proposed countermeasure and invite the community to attempt attacks against the provided benchmarks in the form of a challenge.

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

## APPENDIX A

Table 2 presents the neural network architectures for the victim models used in each benchmark. The network for MNIST is the popular LeNet-3 architecture, the CIFAR-10 architecture is taken from (Ciregan et al. (2012)), and the ImageNet model is inspired by the AlexNet architecture (Krizhevsky et al. (2012)).

Table 2: Baseline (victim) network architectures for evaluated benchmarks. Here, **128C3(2)** denotes a convolutional layer with 128 maps and $3 \times 3$ filters applied with a stride of 2, **MP3(2)** indicates a max-pooling layer over regions of size $3 \times 3$ and stride of 2, and **300FC** is a fully-connected layer consisting of 300 neurons. All convolution and fully connected layers (except the last layer) are followed by ReLU activation. A Softmax activation is applied to the last layer of each network.

| Benchmark | Architecture |
|---|---|
| **MNIST** | $784 - 300FC - 100FC - 10FC$ |
| **CIFAR10** | $3 \times 32 \times 32 - 300C3(1) - MP2(2) - 300C2(1) - MP2(2) -$ 
 $300C3(1) - MP2(2) - 300FC - 100FC - 10FC$ |
| **ImageNet** | $3 \times 224 \times 224 - 96C11(4) - 256C5(1) - MP3(2) - 128C3(1) -$ 
 $MP3(2) - 128C3(1) - 128C3(1) - MP3(2) - 1024FC - 1024FC - 10FC$ |

We visually evaluate the perturbed examples to determine the attack parameters (e.g., perturbation level $\varepsilon$ and $n_{iters}$) such that the perturbations cannot be recognized by a human observer. Table 3 details the parameters used for the realization of different attack algorithms. The JSMA attack for the ImageNet benchmark is computationally expensive (e.g., it took more than 20min to generate one adversarial sample on an NVIDIA TITAN Xp GPU). As such, we could not generate the adversarial samples of this attack using the JSMA library provided by (Nicolas Papernot (2017)).

Table 3: Details of attack algorithms for each evaluated application. The FGS method (Goodfellow et al. (2014)) is characterized with a single $\varepsilon$ parameter. The JSMA attack (Papernot et al. (2016a)) has two parameters: $\gamma$ specifies the maximum percentage of perturbed features and $\theta$ denotes the value added to each selected feature. The Deepfool attack (Moosavi-Dezfooli et al. (2016)) is characterized by the number of iterative updates, which we denote by $n_{iters}$ in this table. For the Carlini&WagnerL2 attack (Carlini & Wagner (2017b)), "C" denotes the confidence, "LR" is the learning rate, "steps" is the number of binary search steps, and "iterations" stands for the maximum number of iterations.

| Application | Attack | Attack Parameters |
|---|---|---|
| MNIST | FGS | $\varepsilon \in \{0.05, 0.1, 0.2, 0.3, 0.4, 0.5\}$ |
| | JSMA | $\gamma = 5\%, \theta \in \{0.3, 0.4, 0.5, 0.6, 0.7, 0.8, 0.9, 1\}$ |
| | Deepfool | $n_{iters} \in \{1, 2, 3, 4, 5, 6, 7, 8, 9, 10\}$ |
| | Carlini&WagnerL2 | $C \in \{0, 0.1, 0.2, 0.3, 0.4, 0.5, 0.6, 0.7, 0.8, 0.9, 0.99\}$ 
 LR = 0.1, steps = 20, iterations = 500 |
| CIFAR | FGS | $\varepsilon \in \{0.05, 0.1, 0.2, 0.3, 0.4, 0.5\}$ |
| | JSMA | $\gamma = 5\%, \theta \in \{0.3, 0.4, 0.5, 0.6, 0.7, 0.8, 0.9, 1\}$ |
| | Deepfool | $n_{iters} \in \{1, 2, 3, 4, 5, 6, 7, 8, 9, 10\}$ |
| | Carlini&WagnerL2 | $C \in \{0, 0.1, 0.2, 0.3, 0.4, 0.5, 0.6, 0.7, 0.8, 0.9, 0.99\}$ 
 LR = 0.1, steps = 20, iterations = 500 |
| ImageNet | FGS | $\varepsilon \in \{0.01, 0.05\}$ |
| | JSMA | Attack not successful |
| | Deepfool | $n_{iters} \in \{1, 2, 3, 4, 5, 6, 7, 8, 9, 10\}$ |
| | Carlini&WagnerL2 | $C \in \{0, 0.1, 0.2, 0.3, 0.4, 0.5, 0.6, 0.7, 0.8, 0.9, 0.99\}$ 
 LR = 0.1, steps = 20, iterations = 500 |

# APPENDIX B

Corresponding ROC curves for PCL performance against FGS, JSMA, Deepfool, and Carlini&WagnerL2 attacks in the CIFAR10 and ImageNet benchmarks.

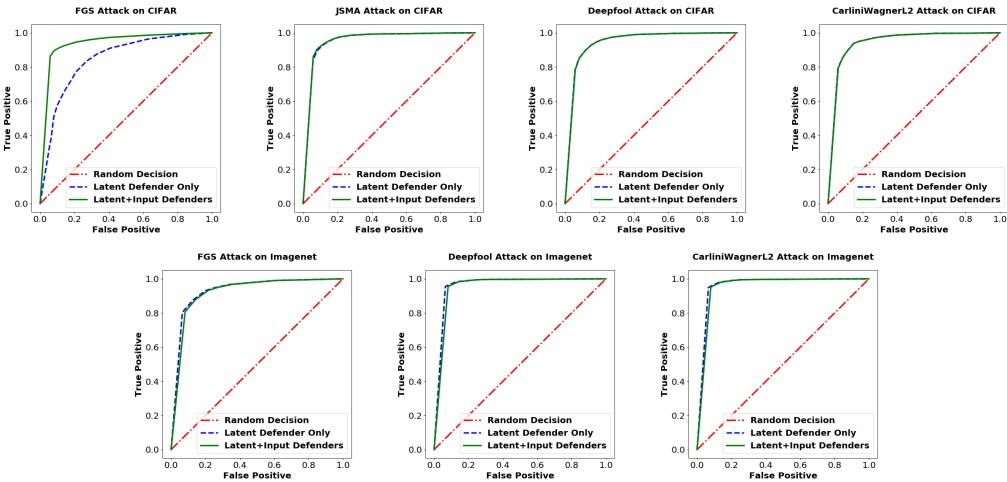

Figure 11: True positive versus false positive rates in CIFAR10 (top row) and ImageNet (bottom row) benchmarks for adversarial samples generated by FGS (Goodfellow et al. (2014)), JSMA (Papernot et al. (2016a)), Deepfool (Moosavi-Dezfooli et al. (2016)), and Carlini&WagnerL2 (Carlini & Wagner (2017b)) attacks.

