# OpenReview forum: "Towards Safe Deep Learning: Unsupervised Defense Against Generic Adversarial Attacks"
_ICLR.cc/2018/Conference — Reject_

### Official Review · AnonReviewer1 · 2017-11-25
**interesting method; difficult to follow**

**Rating:** 5
**Confidence:** 3

**Review:**

This paper present a method for detecting adversarial examples in a deep learning classification setting.  The idea is to characterize the latent feature space (a function of inputs) as observed vs unobserved, and use a module to fit a 'cluster-aware' loss that aims to cluster similar classes tighter in the latent space.

Questions/Comments:

- How is the checkpointing module represented?  Which parameters are fit using the fine-tuning loss described on page 3?

- What is the rationale for setting the gamma (concentration?) parameters to .01?  Is that a general suggestion or a data-set specific recommendation?

- Are the checkpointing modules designed to only detect adversarial examples?  Or is it designed to still classify adversarial examples in a robust way?

Clarity: I had trouble understanding some of this paper.  It would be nice to have a succinct summary of how all of the pieces presented fit together, e.g. the original victim network, fine-tuning loss, per-class dictionary learning w/ OMP.

Technical: It is hard to tell how some of the components of this approach are technically justified.

Novel: I am not familiar enough with adversarial deep learning to assess novelty or impact.

---

> ### Author Response · Authors · 2017-12-21
> **The paper is revised to avoid confusion**
>
> Thank you for your detailed comments. Please find our responses to your questions/comments below.
>
> - Each checkpointing module can be represented as a function \phi(x) along with the centers Ci for different classes. Here, x is the input and \phi(.) is the stack of neural network layers up to the corresponding checkpoint layer. The training of each checkpoint module involves learning \phi() (parameters of the defender neural network) and corresponding centers in the checkpoint layer. Nota that the parameters of the defender module are different than those of the victim model. In other words, each defender module has its own trainable parameters.
>
> - Gamma is a variable among other DL hyperparameters that should be tuned depending on the application data. We empirically found 0.01 to be effective across various benchmarks presented in the paper. However, in general, the user can easily change this hyperparameter in our API depending on her application.
>
> - We have chosen not to use the checkpointing modules as classifiers to focus more on understanding adversarial space, by constructing PDF estimations for legitimate samples. The checkpoint modules are only leveraged to detect adversarial samples that significantly deviate from the underlying PDF.
>
> Clarity:
> Adversarial and legitimate samples differ in certain statistical properties. Adversarial samples are particularly crafted by finding the rarely explored dimensions in a L∞-ball of radius ε. In PCL methodology, samples whose features lie in the unlikely subspaces are identified and rejected as risky samples. Our conjecture is that a general ML model equipped with the side information about the density distribution of the input data as well as the distribution of the latent feature vectors is effectively more robust against adversarial samples.
>
> We formalize the goal of preventing adversarial attacks as an optimization problem to minimize the rarely observed regions in the latent feature space spanned by an ML model. To solve the aforementioned minimization problem, a set of complementary but disjoint checkpoint modules are trained to capture the Probability Density Function (PDF) of the model based on legitimate data samples. The checkpointing modules explicitly characterize the geometry of the input data and the corresponding high-level data abstractions (PDF) within an ML model. In a neural network, for example, each PCL module checkpoints a certain intermediate hidden layer (Figure 1c). The complement of the space characterized by the PDF is marked as the rarely observed region, enabling statistical tests to determine the validity of new samples. Once such characterizations are obtained, statistical testing is used at runtime to determine the legitimacy of new data samples. The defender modules evaluate the input sample probability in parallel with the victim model and raise alarm flags for data points that lie within the rarely explored regions.
>
> The outputs of PCL modules (checkpoints) are aggregated into a single output node (the red neuron in Figure 1c) that quantitatively measures the reliability (confidence) of the victim model prediction. For any input sample, the new neuron outputs a confidence rate in the unit interval [0, 1], with 0 and 1 indicating highly risky and safe samples, respectively. The extra neuron incorporates a “don’t know” class into the model: samples for which the prediction confidence is below a certain threshold are treated as risky inputs. The threshold is determined based on the safety sensitivity of the application for which the ML model is employed.
>
> The per-class dictionaries with OMP reconstruction are further used to assess the Peak Signal to Noise Ratio (PSNR) of incoming samples in the input space and automatically filter out samples that incur low PSNR as demonstrated in Figure 6. Whereas, the latent defenders (PDF) are particularly leveraged to detect adversarial samples with a low perturbation in the input space but high divergence from actual distribution in the latent feature space.

---

### Official Review · AnonReviewer2 · 2017-11-26
**A nice novel idea, but the empirical evaluation is medium and some details are missing in the presentation**

**Rating:** 7
**Confidence:** 3

**Review:**

Summary:
 The paper presents an unsupervised method for detecting adversarial examples of neural networks. The method includes two independent components: an ‘input defender’ which tried to inspect the input, and a ‘latent defender’ trying to inspect a hidden representation. Both are based on the claim that adversarial examples lie outside a certain sub-space occupied by the natural image examples, and modeling this sub-space hence enables their detection. The input defender is based on sparse coding, and the latent defender on modeling the latent activity as a mixture of Gaussians. Experiments are presented on MInst, Cifar10, and ImageNet.

-	Introduction: The motivation for detecting adversarial examples is not stated clearly enough. How can such examples be used by a malicious agent to cause damage to a system? Sketching some such scenarios would help the reader understand why the issue is practically important. I was not convinced it is.
Page 4:
-	Step 3 of the algorithm is not clear:
o	How exactly does HDDA model the data (formally) and how does it estimate the parameters? In the current version, the paper does not explain the HDDA formalism and learning algorithm, which is a main building block in the proposed system (as it provides the density score used for adversarial examples detection). Hence the paper cannot be read as a standalone document. I went on to read the relevant HDDA paper, but it is also not clear which of the model variants presented there is used in this paper.
o	What is the relation between the model learned at stage 2 (the centers c^i) and the model learnt by HDDA? Are they completely different models? Or are the C^I used when learning the HDDA model (and how)?
If these are separate models, how are they used in conjunction to give a final density score? If I understand correctly, only the HDDA model is used to get the final score, and the C^i are only used to make the \phy(x) representation more class-seperable. Is that right?
-	Figure 4, b and c: it is not clear what the (x,y,z) measurements plotted in these 3D drawings are (what are the axis).
Page 5:
-	Section 2: the risk analysis is done in a standard Bayesian way and leads to a ratio of PDFs in equation 5. However, this form is not appropriate for the case presented at this paper, since the method presented only models one of these PDFs (Specifically p(x | W1)  - there is not generative model of p(x|W2)).
-	The authors claim in the last sentence of the section that p(x|W2) is equivalent to 1-p(x|W1), but this is not true: these are two continuous densities, they do not sum to 1, and a model of p(x|W2) is not available (as far as I understand the method)
Page 6:
-	How is equation 7) optimized?
-	Which patchs are extracted from images, for training and at inference time? Are these patchs a dense coverage of the image? Sparsely sampled? Densely sampled with overlaps?
-	Its not clear enough what exactly is the ‘PSNR’ value which is used for the adversarial example detection, and what exactly is ‘profile the PSNR of legitimate samples within each class’. A formal definition of PSNR and’profiling’ is missing (does profiling simply mean finding a threshold for filtering?)
Page 7:
-	Figure 7 is not very informative. Given the ROC curves in figure 8  and table 1 it is redundant.

Page 8:
-	The results in general indicate that the method is much better than chance, but it is not clear if it is practical, because the false alarm rates for high detection are quite high. For example on ImageNet, 14.2% of the innocent images are mistakenly rejected as malicious to get 90% detection rate. I do not think this working point is useful for a real application
-	Given the high flares alarm rate, it is surprising that experiments with multiple checkpoints are not presented (specifically as this case of multiple checkpoints is discussed explicitly in previous sections of the paper).  Experiments with multiple checkpoints are clear required to complete the picture regarding the empirical performance of this method
-	The experiments show that essentially, the latent defenders are stronger than the input defender in most cases. However, an ablation study of the latent defender is missing: Specifcially, it is not clear a) how much does stage 2 (model refinement with clusters)  contribute to the accuracy (how does the model do without it? And 3) how important is the HDDA and the specific variant used (which is not clear) important: is it important to model the Gaussians using a sub-space? Of which dimension?

Overall:
Pros:
-	 A nice idea with some novelty,  based on a non-trivial observation
-	The experimental results how the idea holds some promise
Cons
-	The method is not presented clearly enough: the main component modeling the network activity is not explained (the HDDA module used)
-	The results presented show that the method is probably not suitable for a practical application yet (high false alarm rate for good detection rate)
-	Experimental results are partial: results are not presented for multiple defenders, no ablation experiments


After revision:
Some of my comments were addressed, and some were not.
Specifically, results were presented for multiple defenders and some ablation experiments were highlihgted
Things not addressed:
 - The risk analysis is still not relevant. The authors removed a clearly flawed sentence, but the analysis still assumes that two densities (of 'good' and 'bad' examples) are modeled, while in the work presented only one of them is. Hence this analysis does not add anything to the paper-  it states a general case which does not fit the current scenario and its relation to the work is not clear. It would have been better to omit it and use the space to describe HDDA and the specific variant used in this work, as this is the main tool doing the distinction.

I believe the paper should be accepted.

---

> ### Author Response · Authors · 2017-12-21
> **continued response**
>
> Page 8:
> o We agree with the reviewer that minimizing the false alarm rate is an important concern. As shown through our experiments, the false alarm rate varies in different attack scenarios and benchmarks. We have observed that checkpointing some layers in a deep architecture performs better in terms of minimizing false alarms while achieving a particular detection accuracy. As such, we need to give more weights to those critical checkpoint modules when aggregating the results to whether accept or reject an incoming sample. We modified Section 5 of the paper to include such experiments. In particular, the newly added Figure 10 illustrates the impact of using multiple (three) checkpoints simultaneously in the MNIST benchmark. As demonstrated, using multiple parallel checkpoints with negative correlation can significantly reduce the false positive rate while achieving the same level of detection for adversarial samples.
>
> The space spanned by deep neural networks is far away from being completely understood and we cannot claim to fully understand the space. Our results is a primary step towards a more formal characterization of the adversarial space in the context of deep learning. Investigating the robustness of checkpoint modules at different layers against various attacks is an interesting topic of research. In particular, we are looking into automated techniques to select mixing coefficients for multiple redundancies as future work.
>
> o The cluster refinement is necessary to statistically separate the PDF distributions of adversarial and legitimate samples. The effect of clustering is visualized in Figure 3-b and 3-c, where the histogram of the distance of sample points to the centers (means) of Gaussian Mixture PDFs is shown before and after refinement. The overlap between the two distributions (as in 3-b) injects detection errors: there might be samples that are in fact legitimate, but the detector treats them as adversarial samples. Therefore, the cluster refinement reduces the false alarms.
>
> Please refer to our answer to your second question for details regarding the HDDA method.

---

> ### Author Response · Authors · 2017-12-21
> **Details and evaluations are updated in the revised paper.**
>
> Thank you for your detailed comments. Please find our responses to your questions/comments below.
>
> Introduction:
> For safety-critical applications (e.g., unmanned vehicles and drones), artificial intelligence and machine learning agents will not be trusted until we obtain a better understanding of adversarial space and how to thwart such attacks. In particular,  consider a traffic sign classifier used in self-driving cars. In this example, an adversary can carefully add imperceptible perturbation to a legitimate “stop” sign sample and fool the DL model to classify it as a “yield” sign; thus, jeopardizes the safety of the vehicle. As such, it is highly important to reject risky adversarial samples to ensure the integrity of DL models used in autonomous systems. We modified the second paragraph of the introduction section to further elaborate on the importance of adversarial sample detection.
>
> Page 4:
> o The HDDA algorithm is used in conjunction with the data realignment (step 2 in Figure 2) to learn the density score used for adversarial sample detection. In particular, the data realignment is used to condense data points belonging to each class and HDDA is leveraged to find the corresponding mean and the conditional covariance matrix of a Gaussian Mixture Model that best describe the data points within each class as an ensemble of lower dimensional sub-spaces. We apologize for our over cited reference. We modified Paragraph 1 on page 4 to elaborate more on the HDDA algorithm. Please refer to the slide deck of the cited paper (http://lear.inrialpes.fr/~bouveyron/work/presentation_ASMDA05.pdf) for a succinct summary of the HDDA algorithm (e.g., slide 13).
> -----------------------------------------------------------------------------------
> o Yes, you are right! the Ci is only used for data realignment and make the \phi(x) representation more class-separable. The HDDA is used to learn the mean and conditional covariance matrix of a Gaussian Mixture Model that best describe the data points within each class.
> -----------------------------------------------------------------------------------
> o Thank you for your note. The three axis show the first three Eigenvectors corresponding to the PCA of the data. We used the first three dimensions for visualization purposes only. The data points belong to higher dimensional spaces. We have modified the figure caption to avoid confusion.
>
> Page 5:
> Thank you for your comment. Yes, W1 and W2 are not summed to be 1. we removed the sentence to avoid confusion.
>
> Page 6:
> We modified the 2nd and 3rd Paragraph of Section 3 to address your comments. In particular,  our dictionary learning approach is devised based upon the least angle regression (LAR) method (http://scikit-learn.org/stable/modules/generated/sklearn.decomposition.MiniBatchDictionaryLearning.html) to solve the lasso problem defined in Eq 7. During the training phase, 150,000 randomly selected patches of training data are used to learn the dictionary. Whereas, during the inference phase, all of the non-overlapping patches within the image are denoised by the dictionary to form the whole reconstructed image and compute the PSNR.
>
> We revised the paper (Paragraph 4 of Section 3) to provide the formal definition of PSNR and profiling. In summary, the classic Peak Signal-to-Noise Ratio (PSNR) is defined as PSNR = 20log(MAXI) - 10log(MSE),where the mean square error (MSE) is defined as the L2 difference between the input image and the reconstructed image based on the dictionary (||Zi - DiVi||2). The MAXI is the maximum possible pixel value of the image (usually equivalent to 255). Profiling does mean finding a threshold for filtering; we added the text to the paper to resolve confusion.
>
> Page 7:
> You are right. However, we have included Figure 7 to clearly demonstrate the relation between security parameter, the probability of true detection (True positive), and the probability of false alarm (False positive) for the broad audience and avoid any possible confusion.

---

### Official Review · AnonReviewer3 · 2017-12-02
**An unsupervised manifold projection method for defending adversarial examples based on Gaussian mixture models and dictionary learning. However, similar methodology has been proposed in previous works and the attack evaluations are too weak to claim the contributions.**

**Rating:** 3
**Confidence:** 5

**Review:**

This paper proposes an unsupervised method, called Parallel Checkpointing Learners (PCL), to detect and defend adversarial examples. The main idea is essentially learning the manifold of the data distribution and using Gaussian mixture models (GMMs) and dictionary learning to train a "reformer" (without seeing adversarial examples) to detect and correct adversarial examples. With PCL, one can use hypothesis testing framework to analyze the detection rate and false alarm of different neural networks against adversarial attacks. Although the motivation is well grounded, there are two major issues of this work: (i) limited  novelty - the idea of unsupervised manifold projection method has been proposed in the previous work; and (ii) insufficient attack evaluations - the defender performance is evaluated against weak attacks or attacks with improper parameters. The details are as follows.

1.  Limited novelty and performance comparison - the idea of unsupervised manifold projection method has been proposed and well-studied in "MagNet: a Two-Pronged Defense against Adversarial Examples", appeared in May 2017. Instead of GMMs and dictionary learning in PCL,  MagNet trains autoencoders for defense and provides sufficient experiments to claim its defense capability. On the other hand, the authors of this paper seem to be not aware of this pioneering work and claim "To the best of our knowledge, our proposed PCL methodology is the first unsupervised countermeasure that is able to detect DL adversarial samples generated by the existing state-of-the-art attacks", which is obviously not true. More importantly, MagNet is able to defend the adversarial examples very well (almost 100% success) no matter the adversarial examples are close to the information manifold or not. As a result, the resulting ROC and AUC score are expected be better than PCL. In addition, the authors of MagNet also compared their performance in white-box (attacker knowing the reformer), gray-box (having multiple independent reformers), and black-box (attacker not knowing the reformer) scenarios, whereas this paper only considers the last case.

2. Insufficient attack evaluations - the attacks used in this paper to evaluate the performance of PCL are either weak (no longer state-of-the-art) or incorrectly implemented. For FGSM, the iterative version proposed by (Kurakin, ICLR 2017) should be used. JSMA and deep fool are not considered strong attacks now (see Carlini's bypassing 10 detection methods paper). Carlini-Wagner attack is still strong, but the authors only use 40 iterations (should be at least 500) and setting the confidence=0, which is known to be producing non-transferable adversarial examples. In comparison, MagNet has shown to be effective against different confidence parameters.

In summary, this paper has limited novelty, incremental contributions, and lacks convincing experimental results due to weak attack implementation.

---

> ### Author Response · Authors · 2017-12-21
> **PCL is fundamentally different from manifold projection.  Evaluations cover a wide range of attacks (including the state-of-the-art) to show the generalizability of our methodology.**
>
> Response to comment 1.
>
> Thank you for your comments. However, we strongly disagree with the reviewer’s assessment and conclusion. We have modified the paper (Paragraph 3 of Section 6) to include MagNet as one of the prior work. We would like to highlight the following two points for clarification.
>
> (I) We disagree with the reviewer regarding the robustness of MagNet and particularly the ROC and AUC score of that approach. MagNet and similar works rely on manifold learned by ML agents to “reform” adversarial samples and correct the wrong decision made by the ML agent, e.g., by de-noising samples near the manifold. As shown by Carlini and Wagner in (https://arxiv.org/pdf/1711.08478.pdf), manifold projection methods including MagNet are not robust to adversarial samples. In particular, it has been shown that MagNet only works for very small perturbation values and totally fails (less than 1% detection rate) if the distortion to generate adversarial sample increased by approximately 30% in the worst-case scenario. This performance is even worse than a random prediction (diagonal-line) in our reported ROC curves.
>
> Our proposed PCL methodology is fundamentally different than manifold projection methods such as MagNet. Our conjecture is that the vulnerability of ML models to adversarial samples originates from the existence of rarely explored sub-spaces in each feature map. Due to the curse of dimensionality in modern applications, it is often not practical to fully cover the underlying high-dimensional space spanned by modern ML applications. What we can do, instead, is to construct statistical modules that can quantitatively “detect” whether or not a certain sample comes from the subspaces that were exposed to the ML agent. To ensure robustness against adversarial samples, we argue that ML models should be capable of rejecting samples that are likely to come from the unexplored regions. Unlike manifold projection methods (e.g., MagNet) that particularly rely on data manifolds and remain oblivious to the density of the training samples, our proposed PCL defense methodology learns the probability density function (pdf) of legitimate samples in the latent feature space and raise question marks for risky samples based on the estimated density and an application-specific security parameter. Note that PCL is a detection methodology and is not used to reform data samples.
>
> (II) The ROC curves provided in Figure 8 and Table 1, shows the performance of our proposed PCL methodology in best-case and worst-case scenario by swiping various thresholds that can be selected by a defender depending on the attack and/or application data. As stated in Paragraph 4 of the Introduction section, we assume a white-box attack model in which the attacker knows everything about the victim model including its architecture, learning algorithm, and parameters.
>
> Response to comment 2.
>
> The reviewer is incorrect in the reading of our work. Our aim in including both older and the state-of-the-art attacks is to empirically confirm that our unsupervised PCL methodology can generalize well across a wide range of attacks and is not customized for only one attack scenario. As for implementation, we use the well-known CleverHans adversarial attack library. The reviewer claims our choice of parameters for Carlini’s attack (which the reviewer also admits that this is a strong attack) is not correct. We initially had chosen the default value of the attack based on the original paper. Nevertheless, we have repeated our experiments with the reviewer’s suggested parameters. In particular, for the Carlini attack, we have changed the number of iterations to 500 and created adversarial samples with different confidence parameters in the range 0-99% per your suggestion (updated results are available in Table 1 and Figure 8; new parameter sets can be seen in Table 3). As demonstrated, the performance of our defender is consistent with the results reported previously. This is also consistent with our claim that PCL is rather a generic unsupervised mechanism to detect adversarial samples based on the data distribution in the latent space and is robust against the mechanics of the attack.

---

> > ### Comment · AnonReviewer3 · 2018-01-06
> > **Please take a close read of MagNet and Carlini and Wagner's attack implementation**
> >
> > The reviewer appreciates the authors' efforts in replying the review comments and updating the submitted paper. However, the authors are strongly suggested to take a close read of MagNet and Carlini and Wagner's attack implementation to support the claimed contributions. First of all, MagNet is not simply a manifold projection method. Before entering the reformer for manifold projection, it also uses a detector to reject the adversarial example if its statistical divergence to the training data is large. By adjusting the threshold of the detector in MagNet, it can yield similar ROC and AUC analysis as proposed in this paper. In fact, I think the main difference between this paper and MagNet is in the approach of image reconstruction but not in methodology - MagNet uses auto-encoder and this paper uses dictionary learning. It would be great if we can see the actual defense comparison of these two methods. Secondly, the range of the confidence parameter selected for CW L2 attack in Table is not representative. Based on the evaluation of their S&P paper, the confidence should be selected from the range between 0 to 60 (or 100) instead of 0 to 1 reported in this paper.

---

### Decision · Program_Chairs · 2018-01-29
**ICLR 2018 Conference Acceptance Decision**

**Decision:**

Reject

**Comment:**

The paper proposes a method to detect and correct adversarial examples at the input stage (using a sparse coding based model) and/or at a hidden layer (using a GMM). These detector/corrector models are trained using only the natural examples. While the proposed method is interesting and has some novelty wrt to the specific models used for detection/correction (ie sparse coding and GMMs), there are crucial gaps in the empirical studies:

- It does not compare with a highly relevant prior work MagNet (Meng and Chen, 2017) which also detects and corrects adversarial examples by modeling the distribution of the natural examples

- The attacks used in the evaluations do not consider the setting where the existence (and architecture) of the defender models is known to the attacker

- It does not evaluate the method on a stronger PGD attack (also known as iterative FGSM)